

# Pitfalls associated with evaluating enzymatic quorum quenching activity: the case of MomL and its effect on *Pseudomonas aeruginosa* and *Acinetobacter baumannii* biofilms

Yunhui Zhang, Gilles Brackman and Tom Coenye

Laboratory of Pharmaceutical Microbiology, Ghent University, Gent, Belgium

## ABSTRACT

**Background.** The enzymatic degradation of quorums sensing (QS) molecules (called quorum quenching, QQ) has been considered as a promising anti-virulence therapy to treat biofilm-related infections and combat antibiotic resistance. The recently-discovered QQ enzyme MomL has been reported to efficiently degrade different $N$-acyl homoserine lactones (AHLs) of various Gram-negative pathogens. Here we investigated the effect of MomL on biofilms formed by two important nosocomial pathogens, *Pseudomonas aeruginosa* and *Acinetobacter baumannii*.

**Methods.** MomL was expressed in *E.coli* BL21 and purified. The activity of MomL on AHLs with hydroxyl substituent was tested. Biofilms of *P. aeruginosa* PAO1 and *Acinetobacter* strains were formed in 96-well microtiter plates. Biofilm formation was evaluated by crystal violet staining, plating and fluorescence microscopy. The effect of MomL on biofilm susceptibility to antibiotics was also tested. We further evaluated MomL in dual-species biofilms formed by *P. aeruginosa* and *A. baumannii*, and in biofilms formed in a wound model. The effect of MomL on virulence of *A. baumannii* was also tested in the *Caenorhabditis elegans* model.

**Results.** MomL reduced biofilm formation and increased biofilm susceptibility to different antibiotics in biofilms of *P. aeruginosa* PAO1 and *A. baumannii* LMG 10531 formed in microtiter plates *in vitro*. However, no significant differences were detected in the dual-species biofilm and in wound model biofilms. In addition, MomL did not affect virulence of *A. baumannii* in the *C. elegans* model. Finally, the effect of MomL on biofilm of *Acinetobacter* strains seems to be strain-dependent.

**Discussion.** Our results indicate that although MomL showed a promising anti-biofilm effect against *P. aeruginosa* and *A. baumanii* biofilms formed in microtiter plates, the effect on biofilm formation under conditions more likely to mimic the real-life situation was much less pronounced or even absent. Our data indicate that in order to obtain a better picture of potential applicability of QQ enzymes for the treatment of biofilm-related infections, more elaborate model systems need to be used.

Corresponding author
Tom Coenye,
Tom.Coenye@UGent.be

## INTRODUCTION

Quorum sensing (QS) is a widespread communication process that allows bacteria to coordinate their group behavior based on the production, detection and response to extracellular signal molecules (*Bassler & Losick, 2006*; *Williams et al., 2007*). QS regulates gene expression related to biofilm formation, motility and production of virulence factors in many Gram-negative and Gram-positive pathogens, and interfering with QS has been intensively studied as a promising anti-virulence therapy to combat bacterial infections and antibiotic resistance (*Hentzer & Givskov, 2003*; *LaSarre & Federle, 2013*; *Rutherford & Bassler, 2012*). Many natural and synthetic compounds have been found to inhibit QS, and several quorum quenching (QQ) enzymes mainly targeting $N$-acyl homoserine lactone (AHL) based QS in Gram-negative bacteria have been described as well (*Brackman & Coenye, 2015*; *Fetzner, 2015*; *Rasmussen & Givskov, 2006*; *Tang & Zhang, 2014*). Some of these QS inhibitors (QSIs) and QQ enzymes have shown promising anti-virulence effects both *in vitro* and *in vivo*. For instance, furanones which resemble AHLs and are able to bind to QS receptors have been reported to reduce biofilm formation and enhance bacterial clearance in *Pseudomonas aeruginosa* lung infection in mice (*Hentzer et al., 2002*; *Wu et al., 2004*). Baicalin hydrate and cinnamaldehyde (QSIs targeting AHL-based QS in *P. aeruginosa* and *Burkholderia cepacia* complex) as well as hamamelitannin (a QSI targeting the peptide-based QS system present in *Staphylococcus aureus*) increase biofilm susceptibility to antibiotics and survival of infected *Galleria mellonella* larvae and *Caenorhabditis elegans*, as well as decrease the microbial load in a mouse pulmonary infection model (*Brackman et al., 2011*). As for QQ enzymes, an AiiM-producing *P. aeruginosa* mutant showed reduced lung injury and increased survival in an *in vivo* study on mice with pneumonia (*Migiyama et al., 2013*), and an inhaled lactonase *Sso*Pox-I was also reported to reduce virulence of *P. aeruginosa* and mortality in rat pneumonia (*Hraiech et al., 2014*).

Previously MomL, a novel AHL lactonase belonging to the metallo-$\beta$-lactamase superfamily was identified and characterized (*Tang et al., 2015*). It has high degrading activities towards short- and long-chain AHLs with or without an oxo-group at the C-3 position (*Tang et al., 2015*). MomL can reduce pyocyanin and protease production by *P. aeruginosa* and attenuated the virulence of *P. aeruginosa* in a *C. elegans* infection model (*Tang et al., 2015*), but its effect on biofilm formation of *P. aeruginosa* and other Gram-negative pathogens was not tested yet.

Besides *P. aeruginosa*, *Acinetobacter baumannii* has also been recognized as an increasingly prevalent Gram-negative opportunistic pathogen responsible for severe nosocomial infections (*Gonzalez-Villoria & Valverde-Garduno, 2016*; *Peleg, Seifert & Paterson, 2008*). Resistance of *P. aeruginosa* and *A. baumannii* strains to multiple antibiotic classes complicates the treatment for these infections and poses considerable therapeutic challenges worldwide (*Potron, Poirel & Nordmann, 2015*). One of the main factors contributing to their reduced antibiotic susceptibility and to treatment failure is biofilm formation both on tissues and abiotic surfaces (*Donlan & Costerton, 2002*; *Hall-Stoodley, Costerton & Stoodley, 2004*; *Longo, Vuotto & Donelli, 2014*). Biofilms of both *P. aeruginosa* and *A. baumannii*

are known to be regulated by AHL-based QS. In *P. aeruginosa*, *N*-(3-oxododecanoyl)-L-homoserine lactone (3-oxo-C12-HSL) and *N*-butyryl-L-homoserine lactone (C4-HSL) are used by the Las and Rhl QS system, respectively (*Pesci et al., 1997*), and these AHLs can both be degraded by MomL (*Tang et al., 2015*). One AHL synthase belonging to LuxI family, AbaI, has been reported to catalyze the synthesis of *N*-(3-hydroxydodecanoyl)-L-HSL (3-OH-C12-HSL) in *Acinetobacter nosocomialis* M2 (*Niu et al., 2008*), but other AHLs with varied chain lengths and substituents are also found in *Acinetobacter* strains (*Bhargava, Sharma & Capalash, 2010*; *González et al., 2009*). The biofilm-forming ability of an *abaI* mutant was reduced by around 40% compared to the corresponding wildtype strain (*Niu et al., 2008*). Compared to the extensive literature on inhibiting QS pathways and virulence in *P. aeruginosa* (*Aybey & Demirkan, 2016*; *Furiga et al., 2016*; *Hentzer et al., 2003*; *O'Loughlin et al., 2013*; *Yin et al., 2015*), there are fewer reports on inhibiting QS and biofilm formation in *A. baumannii* (*Bhargava et al., 2015*; *Chow et al., 2014*; *Saroj & Rather, 2013*).

In the present study, we tested the anti-biofilm activity of MomL against *P. aeruginosa* and *A. baumannii*, and further evaluated the effect of MomL under more complex conditions such as in dual-species biofilm and in a wound model system with the aim to obtain a better knowledge base regarding the possible development of QQ enzymes as anti-virulence therapy.

## MATERIAL & METHODS

### Bacterial strains, culture conditions and chemicals

*P. aeruginosa* PAO1, *A. calcoaceticus* LMG 10517, *A. nosocomialis* M2 and *A. baumannii* LMG 10520, LMG 10531 and AB5075 were cultured on tryptic soy agar (TSA) or in Mueller-Hinton broth (MH) at 37 °C. *Escherichia coli* BL21(DE3) harboring MomL expression plasmid pET24a(+)-momL-(−SP) (*Tang et al., 2015*) was cultured on Luria-Bertani (LB) agar supplemented with kanamycin (50 µg/mL) at 37 °C. The AHL biosensor *Agrobacterium tumefaciens* A136 (pCF218) (pCF372) (*Zhu et al., 1998*) was maintained on LB agar supplemented with spectinomycin (50 µg/mL) and tetracycline (4.5 µg/mL), and grown in AT minimal medium (*Tempé et al., 1977*) containing 0.5% (wt/vol) glucose for detecting AHLs in the liquid X-Gal (5-bromo-4-chloro-3-indolyl- $\beta$-D-galactopyranoside) assay. 3-OH-C12-HSL was purchased from Sigma-Aldrich and dissolved in dimethyl sulfoxide (DMSO) as stock solution (100 mM). *Caenorhabditis elegans* N2 (glp-4; sek-1) was propagated under standard conditions, synchronized by hypochlorite bleaching, and cultured on nematode growth medium using *E. coli* OP50 as a food source (*Stiernagle, 1999*).

### MomL expression and purification

MomL was expressed and purified according to *Tang et al. (2015)*. In brief, protein expression was induced by 0.5 mM IPTG (isopropyl- β-D-thiogalactopyranoside) when *E. coli* cells in LB reaching an optical density at 600 nm (OD600) of 0.5–0.7. The induction was carried out at 16 °C with moderate shaking (150 rpm) for 12 h. Cells were harvested and sonicated, and the obtained supernatant was loaded on NTA-Ni (Qiagen, Hilden, Germany) columns for purification according to the manufacturer's instruction. Desalting of the protein solution was accomplished by Amicon Ultra-15 centrifugal filter devices,

and the purified MomL was stored at −20 °C in Tris–HCl buffer (50 mM, pH 6.5) with 25% glycerol.

## Detection for degradation of 3-OH-C12-HSL

The amount of 3-OH-C12-HSL was quantified using *A. tumefaciens* A136 liquid X-gal assay and expressed as the normalized $\beta$-galactosidase activity as previously described (*Tang et al., 2013*). The correction factor a and b were obtained and calculated for our experimental conditions, and the final formula to calculate the normalized $\beta$-galactosidase activity is $\frac{0.716 \times \ OD492 - OD620}{0.205 \times \ OD620 - OD492}$. To test the degradation of 3-OH-C12-HSL by MomL, 3-OH-C12-HSL (10 µM) was mixed with MomL in different concentrations (0.05–5 µg/mL) and incubated at 37 °C for 1 h. No MomL was added to the control. Afterwards the residual 3-OH-C12-HSL was quantified by adding 10 µL solution to the A136 biosensor, as described previously (*Tang et al., 2013*).

## Biofilm formation assays

Overnight cultures of *P. aeruginosa* and *Acinetobacter* strains in MH broth were diluted to contain approximately $5 \times 10^7$ CFU/mL. 90 µL of this diluted bacterial suspension was transferred to the wells of a round-bottomed 96-well microtiter plate. Uninoculated MH broth served as blank control. To test the effect of MomL on biofilms, 10 µL purified enzyme (in different concentrations) was added to the wells, while 10 µL Tris–HCl buffer (50 mM, pH 6.5) with 25% glycerol was added to the control. The plate was incubated at 37 °C for 4 h before the supernatant was removed. The wells were rinsed once with sterile physiological saline (PS) and re-filled with fresh medium (90 µL) and MomL (10 µL). The plate was incubated at 37 °C for an additional 20 h. The biofilm biomass was quantified by crystal violet (CV) staining as described previously (*Peeters, Nelis & Coenye, 2008*). After rinsing the wells with sterile PS, the biofilm was fixed with 100 µL 99% methanol for 15 min and stained with 100 µL 0.1% CV for 20 min. The excess CV was removed by washing the plates under running tap water and bound CV was released by adding 150 µl of 33% acetic acid. The absorbance was measured at 590 nm.

## Biofilm susceptibility assays

After a 24 h-biofilm of *P. aeruginosa* or *A. baumannii* strains was formed as described above either in presence of MomL or not, the plate was emptied and biofilm cells were rinsed with sterile PS. Antibiotics were dissolved in PS and 90 µL of these solutions were added to treat the biofilm for another 24 h, either with or without 10 µl MomL. Tobramycin (TOB; 4 µg/mL as final concentration), ciprofloxacin (CIP; 0.5 µg/mL), meropenem (MEM; 16 µg/mL) and colistin (CST; 16 µg/mL) were used to treat the biofilm of *P. aeruginosa* PAO1; TOB (6 µg/mL), CIP (4 µg/mL), MEM (8 µg/mL) and CST (16 µg/mL) were used to treat the biofilm of the *A. baumannii* strains. The supernatant was removed and the wells were washed once with sterile PS. To release bacterial cells from biofilm, two cycles of vortex (5 min) and sonication (5 min) were performed, and the number of CFU/biofilm was determined by plating the resulting suspensions on TSA.

## Fluorescence microscopy

Biofilms of *P. aeruginosa* PAO1 or *A. baumannii* strains were formed in the absence or presence of MomL and treated with antibiotics as described above using a flat-bottomed 96-well microtiter plates. A total of 3 μL SYTO9 and 3 μL propidium iodide were diluted to 1mL in sterile PS, and 100 μL of this staining solution was transferred to each well. The plate was incubated for 15 min at room temperature and fluorescence microscopy was performed with EVOS FL Auto Imaging System (Life Technologies, Carlsbad, CA, USA). The red fluorescent signal was detected with 531/40 nm excitation filter cube and 593/40 nm emission filter cube and the green fluorescent signal was detected with 470/22 nm excitation filter cube and 510/42 nm emission filter cube.

## Dual-species biofilm formation

Overnight cultures of *P. aeruginosa* and *A. baumannii* strains in MH broth were diluted to contain approximately $5 \times 10^5$ CFU/mL and $5 \times 10^7$ CFU/mL, respectively, and equal volume of suspensions of *P. aeruginosa* and *A. baumannii* were mixed. MomL (200 μg/mL) and tobramycin (6 μg/mL) were added as described above. To quantify CFU in the dual-species biofilm, *Pseudomonas* Isolation Agar (Difco) and TSA supplemented with 5 μg/mL cefsulodin were used as selective media for *P. aeruginosa* and *A. baumannii* respectively.

## Biofilm formation in wound model

Artificial dermis composed of hyaluronic acid and collagen was used in our wound model, as described before (*Brackman et al., 2016*). Each disk of artificial dermis was placed in 24-well microtiter plate. One mL medium containing Bolton Broth, heparinized bovine plasma and freeze-thaw laked horse blood cells was added on and around the dermis. Suspensions (10 μL) of *P. aeruginosa* or *A. baumannii* containing $10^4$ bacterial cells were added on the top of dermis. Final concentrations of MomL added were 200 μg/mL and 10 μg/mL for *P. aeruginosa* and *A. baumannii*, respectively. Tobramycin (10 μg/mL) was added after 8 h incubation at 37 °C. After 24 h, the infected dermis was washed with 1 mL PS and was transferred into 10 ml PS. Biofilm cells on the dermis were loosen and collected by three cycles of vortex (30 s) and sonication (30 s). The number of CFU/dermis was quantified by standard plating techniques.

## *C. elegans* survival assay

The *C. elegans* survival assay was carried out as described before with minor modification (*Brackman et al., 2011*). Synchronized worms (L4 stage) were suspended in medium containing 95% M9 buffer (3 g of $KH_2PO_4$, 6 g of $Na_2HPO_4$, 5 g of NaCl, and 1 ml of 1 M $MgSO_4 \cdot 7H_2O$ in 1 l of water) and 5% brain heart infusion broth (Oxoid), and 25 μL of this nematode suspension was transferred to the wells of a 96-well microtiter plate. Overnight culture of *A. baumannii* was suspended in the assay medium and added in a final concentration of $2.5 \times 10^7$ CFU/ml. MomL was added in a final concentration of 10 μg/mL. The plates were incubated at 25 °C for 24 h. The fraction of dead worms was determined by counting the number of dead worms and the total number of worms in each well.

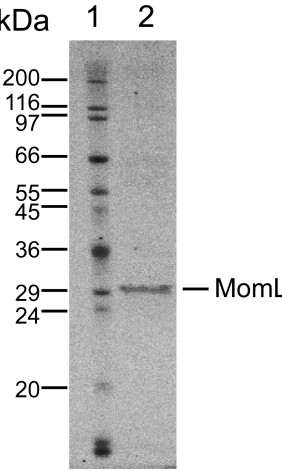

**Figure 1** **SDS-PAGE of purified MomL.** Lane1, molecular mass markers; Lane 2, purified recombinant MomL with molecular mass of nearly 31 kDa.

## Statistics

The normal distribution of the data was checked by the D'Agostino–Pearson normality test. Normally distributed data were analyzed by one-way ANOVA, and non-normally distributed data were analyzed by the Kruskal–Wallis test or the Mann–Whitney test. All statistical analyses were carried out using GraphPad Prism 6.0.

## RESULTS

### Degradation of 3-OH-C12-HSL by purified MomL

MomL was produced in *E. coli* and subsequently successfully purified (Fig. 1). Although MomL had been shown to degrade various AHLs (*Tang et al., 2015*), its activity on AHLs with hydroxyl substituent at the C3 position was not tested yet. We could demonstrate that MomL, in a concentration of 1 μg/mL or higher, can degrade almost all 3-OH-C12-HSLs (10 μM) under the experimental conditions used in the present study (Fig. 2).

### Effect of MomL on biofilm formation by *P. aeruginosa* and *A. baumannii*

Following biofilm formation in 96-well microtiter plates and quantification by crystal violet staining, a significant difference was observed between *P. aeruginosa* PAO1 control biofilms and biofilms grown in the presence of MomL (concentration > 50 μg/mL) (Fig. 3A). When grown with 150 μg/mL MomL, an average decrease of approximately 35% was observed. MomL inhibited *A. baumannii* LMG 10531 biofilm formation at concentrations as low as 0.1 μg/mL, and the biofilm biomass was reduced by approximately 42% when exposed to 5 μg/mL MomL (Fig. 3B). No further decrease was observed when *A. baumannii* LMG 10531 biofilms were grown in the presence of higher concentration of MomL.

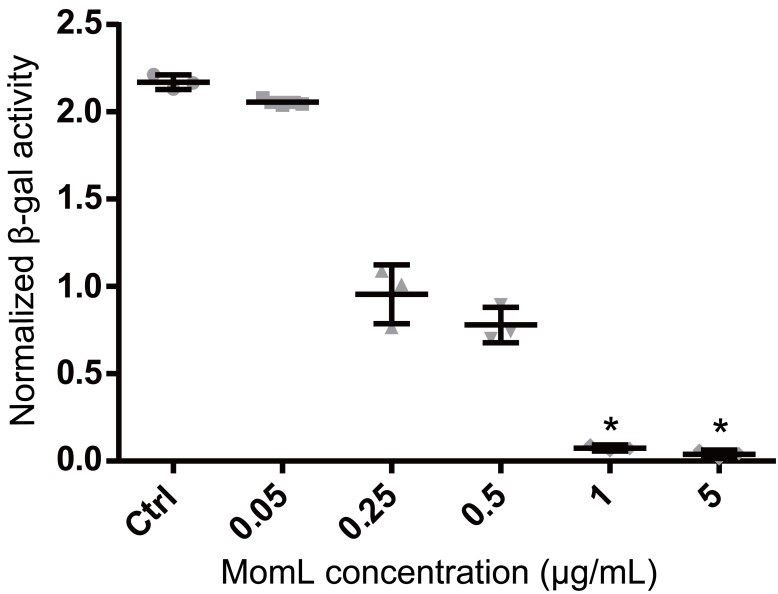

**Figure 2** **Degradation of 3OH-C$_{12}$-HSL by MomL.** The amount of residual 3-OH-C12-HSL was expressed as the normalized $\beta$-galactosidase activity. Data shown are average of three technical replicates ($n = 3$), error bars represent standard deviation. *, $P < 0.05$ when compared with non-MomL treated control (Kruskal-Wallis test).

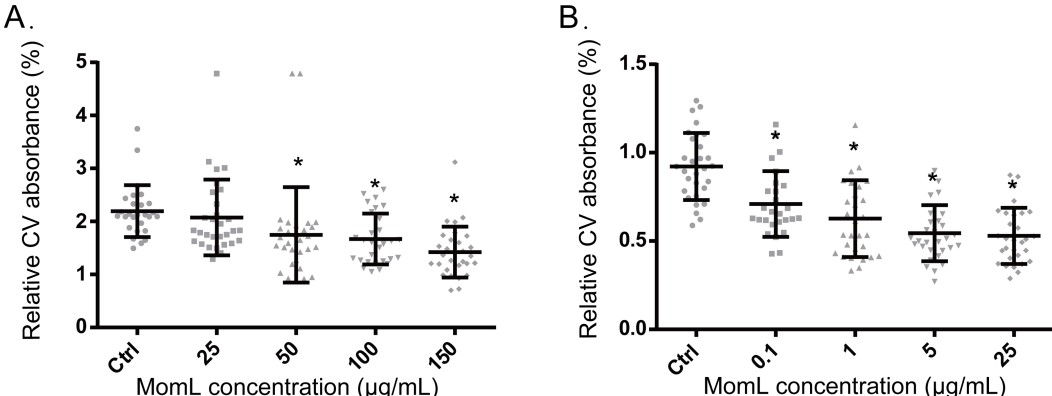

**Figure 3** **Effect of MomL on biofilm formation by *P. aeruginosa* PAO1 (A) and *A. baumannii* LMG 10531 (B).** Biofilms were quantified by CV staining and amount of biofilm left is expressed as CV absorbance (OD 590). Data shown are average of three biological replicates with variable numbers of technical replicates each ($n \geq 27$), error bars represent standard deviation. *, $P < 0.05$ when compared with non-MomL treated control in Kruskal-Wallis test (A) or one-way ANOVA(B).

## Effect of MomL on biofilm susceptibility to antibiotics

Application of MomL alone (200 µg/mL for *P. aeruginosa* PAO1 and 10 µg/mL for *A. baumannii* LMG 10531) reduced the number of cultivable biofilm cells by approximately 50% in both *P. aeruginosa* PAO1 and *A. baumannii* LMG 10531. For *P. aeruginosa* PAO1, combining CIP or MEM with MomL led to >70% more reduction compared to treatment with CIP or MEM alone (Fig. 4A). For *A. baumannii* LMG 10531, MomL also increased killing of biofilm cells when antibiotics were used together with MomL (Fig. 4B). In case of

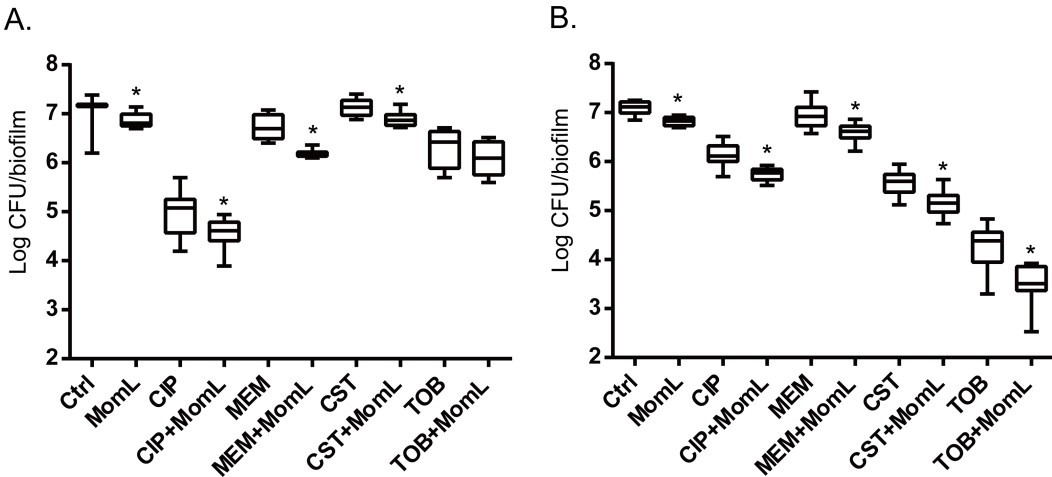

**Figure 4** **Effect of MomL on susceptibility of *P. aeruginosa* PAO1 (A) and *A. baumannii* LMG 10531 (B) biofilms to different antibiotics.** Numbers of CFU/biofilm were determined by plating and shown as box-whisker plots. Boxes span the interquartile range; the line within each box denotes the median, and whiskers indicate the minimum and maximum values. MomL was added in a final concentration of 200 μg/mL for *P. aeruginosa* PAO1 and 10 μg/mL for *A. baumannii* LMG 10531. Data shown are from three biological replicates with three technical replicates each ($n = 9$). Mann–Whitney tests were performed to compare different groups (*, $P < 0.05$)

TOB, cell number was reduced by 80% when used in combination with MomL compared to TOB alone. Consistent with results obtained by plating, fewer living cells were observed in fluorescence microscope images of biofilms treated with MomL, TOB, or a combination of both, compared to control biofilms (Fig. 5).

## Effect of MomL on dual-species biofilm formed by *P. aeruginosa* and *A. baumannii*

We also evaluated the effect of MomL on dual-species biofilm formed by *P. aeruginosa* PAO1 and *A. baumannii* LMG 10531. We found that *P. aeruginosa* PAO1 inhibited growth of *A. baumannii* LMG 10531 in dual-species biofilm, and most *A. baumannii* LMG 10531 cells were killed by *P. aeruginosa* PAO1 after 48 h (Fig. 6). When MomL was added, there was a reduction in *A. baumannii* LMG 10531 cell numbers; however no difference was observed in either total cell numbers or number of surviving *P. aeruginosa* PAO1 cells (Fig. 6A). MomL in combination of TOB was also tested, but no change in susceptibility to TOB was observed in the dual-species biofilm (Fig. 6B).

## Effect of MomL on other *Acinetobacter* strains

We also tested MomL on four other *Acinetobacter* strains. However, only *A. baumannii* LMG 10520 showed reduction in biofilm biomass when treated with MomL at 50 μg/mL (Fig. 7). No significant difference was observed for *A. calcoaceticus* LMG 10517, *A. nosocomialis* M2 and *A. baumannii* AB5075. The effect of MomL on susceptibility of *A. baumannii* LMG 10520 and *A. calcoaceticus* LMG 10517 biofilms was also tested. For *A. baumannii* LMG 10520, significant differences were detected when MomL was added alone or in combination with several antibiotics (Fig. 8). For *A. calcoaceticus* LMG 10517, no difference

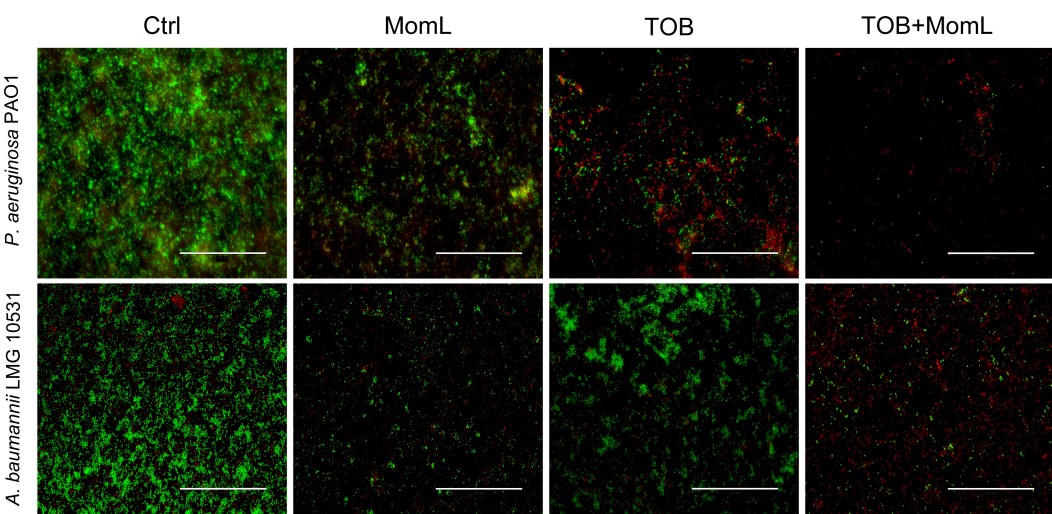

**Figure 5** **Representative fluorescence images of biofilms of *P. aeruginosa* PAO1 and *A. baumannii* LMG 10531.** Biofilms were treated with MomL alone, TOB alone or a combination of both and stained with Syto9 and propidium iodide. 40× Objective (numerical aperture: 0.75) was used and the final magnification is 1,200×. The scale bar represents 100 μm.

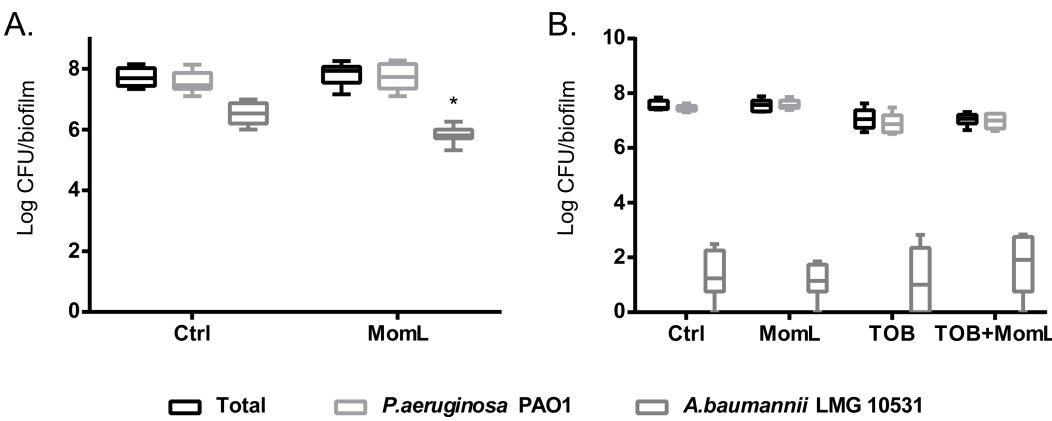

**Figure 6** **Effect of MomL on dual-species biofilms.** Total number of CFU/biofilm, number of *P. aeruginosa* PAO1 CFU/biofilm and number of *A. baumannii* LMG 10531 CFU/biofilm in each dual-species biofilm were determined by plating and shown as box-whisker plots. Boxes span the interquartile range; the line within each box denotes the median, and whiskers indicate the minimum and maximum values. (A). 24 h-biofilm treated with MomL alone; (B). 48 h-biofilm treated with MomL alone, TOB alone or a combination of both. Data shown are from three biological replicates with three (A) or two (B) technical replicates each ($n = 9$ for A, $n = 6$ for B). Mann–Whitney tests were performed to compare total, *P. aeruginosa* PAO1 and *A. baumannii* LMG 10531 cell numbers respectively between untreated or MomL-treated dual-species biofilm (*, $P < 0.05$).

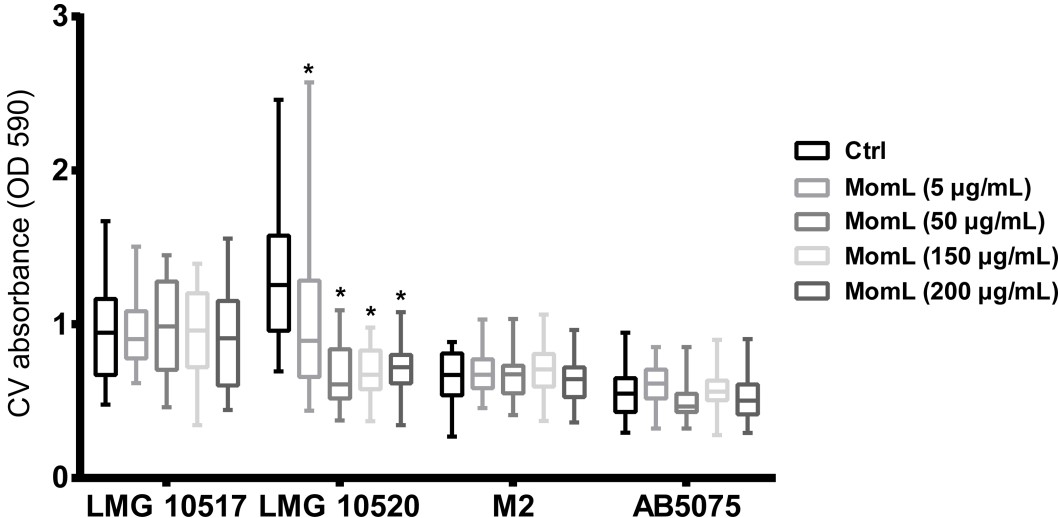

**Figure 7** **Effect of MomL on biofilms formed by other *Acinetobacter* strains.** Biofilms of *A. calcoaceticus* LMG 10517, *A. nosocomialis* M2, *A. baumannii* LMG 10520 and *A. baumannii* AB5075 were treated with different concentration of MomL and quantified by CV staining. Data shown in box-whisker plots are from three biological replicates with variable numbers of technical replicates each ($n \geq 27$). Boxes span the interquartile range; the line within each box denotes the median, and whiskers indicate the minimum and maximum values. *, $P < 0.05$ when compared to untreated control (Kruskal-Wallis test).

was observed between biofilms receiving MomL treatment and biofilms receiving the control treatment, either by plating or fluorescence microscope.

### Effect of MomL in a biofilm wound model system and in the *C. elegans* model

An *in vitro* wound model was used to mimic the conditions in an infected wound. For both *P. aeruginosa* PAO1 and *A. baumannii* LMG 10531, MomL had no effect on biofilm formation in this wound model (Fig. 9).

The *C. elegans* model was used to further evaluate whether MomL can increase survival of nematodes infected with *A. baumannii*. However, no significant increase of *C. elegans* survival was found after treating nematodes infected with *A. baumannii* LMG 10520 or *A. baumannii* LMG 10531 with MomL (Fig. 10), although AHL-degrading activity was maintained under these test conditions (Fig. S2).

## DISCUSSION

QS disruption has been considered as a promising anti-infectious strategy to substitute or at least supplement treatment with antibiotics, and could inhibit production of virulence factors and the formation of biofilms (*Brackman et al., 2011*). Compared to QS inhibitors, QQ enzymes can degrade AHLs from different pathogens and might be more effective in treating multispecies infections. In addition, QQ enzymes do not need to enter the cells as they can act extracellularly, making it less likely resistance will develop (*Bzdrenga et al., 2016*). The recently-discovered QQ enzyme, MomL, has strong degrading activity towards AHLs with different acyl-chain length and substituents (oxo or hydroxyl) (*Tang et al., 2015*),

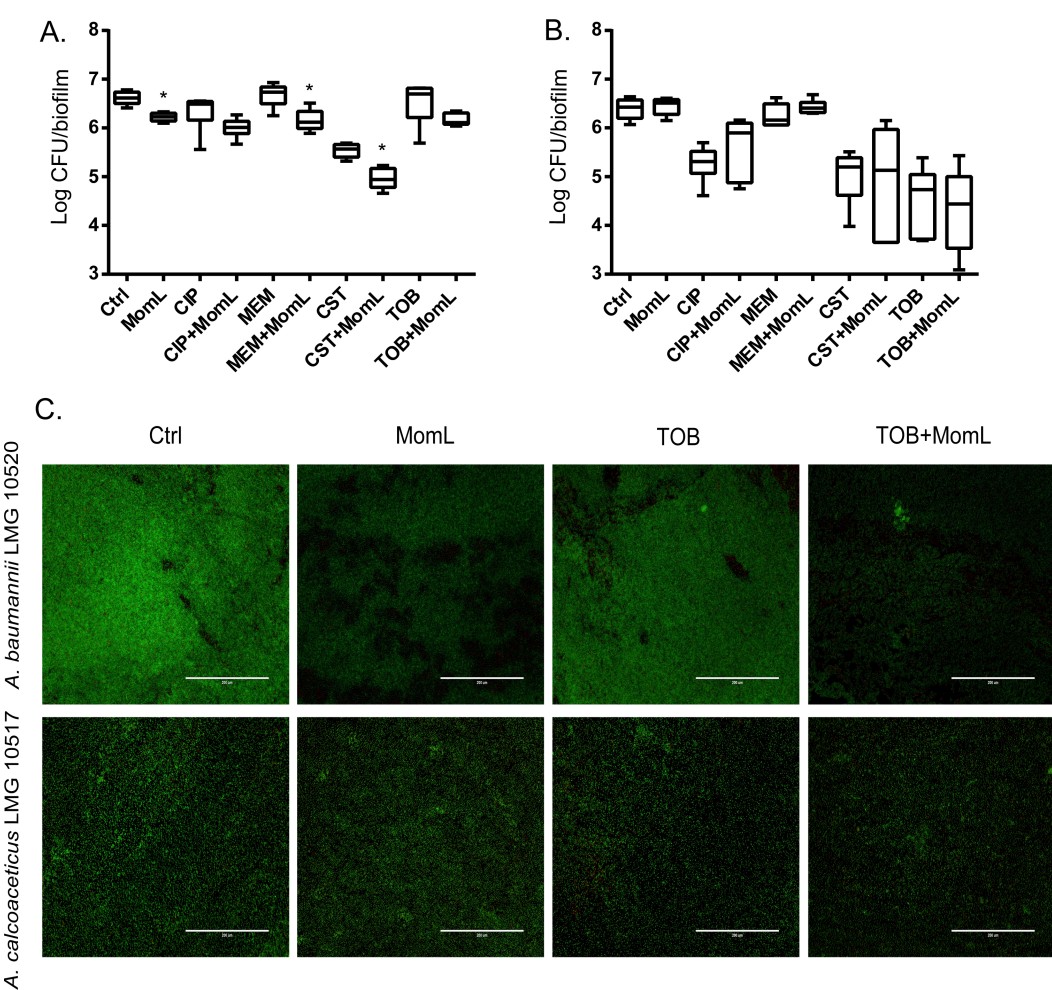

**Figure 8** **Effect of MomL on biofilm susceptibility of *A. baumannii* LMG 10520 and *A. calcoaceticus* LMG 10517.** (A). Plating results for biofilms of *A. baumannii* LMG 10520 exposed to CIP, MEM, CST, TOB alone or in combination with MomL (50 μg/mL); (B), Plating results for biofilms of *A. calcoaceticus* LMG 10517 exposed to CIP, MEM, CST, TOB alone or in combination with MomL (200 μg/mL). Data shown in box-whisker plots are from two biological replicates with three technical replicates each ($n = 6$). Boxes span the interquartile range; the line within each box denotes the median, and whiskers indicate the minimum and maximum values. Mann–Whitney tests were performed to compare control and MomL or antibiotic treatment alone and in combination with MomL ($*$, $P < 0.05$). (C). Representative fluorescence images of *A. baumannii* LMG 10520 and *A. calcoaceticus* LMG 10517. Biofilms were treated with MomL alone or in combination with tobramycin and stained with Syto9 and propidium iodide. 20× Objective (numerical aperture: 0.65) was used and the final magnification is 599×. The scale bar represents 200 μm.

and this could be an advantage when targeting bacteria like *Acinetobacter* strains that produce various AHLs. MomL was reported to reduce the *in vitro* production of protease and pyocyanin by *P. aeruginosa* and attenuate the virulence of *P. aeruginosa* in a *C. elegans* infection model (*Tang et al., 2015*). The further application potential of MomL was not determined yet. In the present study we investigated the possible use of MomL for treating biofilm infections, and evaluate its effect on two important Gram-negative nosocomial pathogens, *P. aeruginosa* and *A. baumannii* in different models.

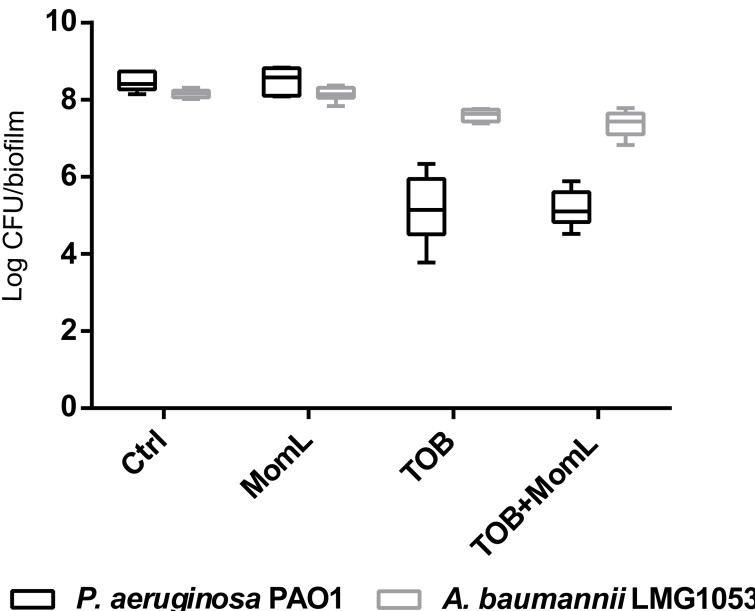

**Figure 9   Effect of MomL on biofilms of *P. aeruginosa* PAO1 and *A. baumannii* LMG 10531 formed in wound model.** Data shown in box-whisker plots are from three biological replicates with two technical replicates each ($n = 6$). Boxes span the interquartile range; the line within each box denotes the median, and whiskers indicate the minimum and maximum values. Mann–Whitney tests were performed to compare control and MomL treatment, or TOB and TOB in combination with MomL.

First we tested the effect of MomL on single-species biofilms of *P. aeruginosa* PAO1 and *A. baumannii* LMG 10531 formed in microtiter plates; a reduction of biofilm biomass was observed for both strains. The maximum decrease in biofilm of *A. baumannii* LMG 10531 was achieved at a concentration of 5 µg/mL and no further decrease was observed with higher concentrations of MomL, which indicated the presence of other mechanisms besides QS regulating biofilm formation in *A. baumannii*. Higher concentrations of MomL are needed to show an effect on biofilm of *P. aeruginosa* PAO1 comparing to *A. baumannii* LMG 10531 (Fig. 2), possibly due to the higher concentration of AHLs produced by PAO1. When used in combination with antibiotics, fewer biofilm cells survived compared to antibiotic treatment alone, both for *P. aeruginosa* PAO1 and *A. baumannii* LMG 10531. In addition, MomL showed no inhibition on planktonic cells of both *P. aeruginosa* and *A. baumannii* (Fig. S1), and all these *in vitro* results seem promising and suggest possible use of MomL to treat biofilm infections of *P. aeruginosa* and *A. baumannii*.

We subsequently investigated the effect of MomL in a dual-species biofilm formed by *P. aeruginosa* and *A. baumannii* and in a wound biofilm model. MomL had no effect on the overall cell number in the mixed species biofilm and the same disappointing results were obtained in biofilms formed in wound model system. In this wound model system, medium containing plasma, serum, horse blood and heparin were used to reflect nutritional condition in wounds. An artificial dermis was used to mimic a wound like surface and an inoculum of $10^4$ cells was used to reflect the microbial load of a wound prior to infection. Additionally, in contrast to what we observed for the mono-species biofilms

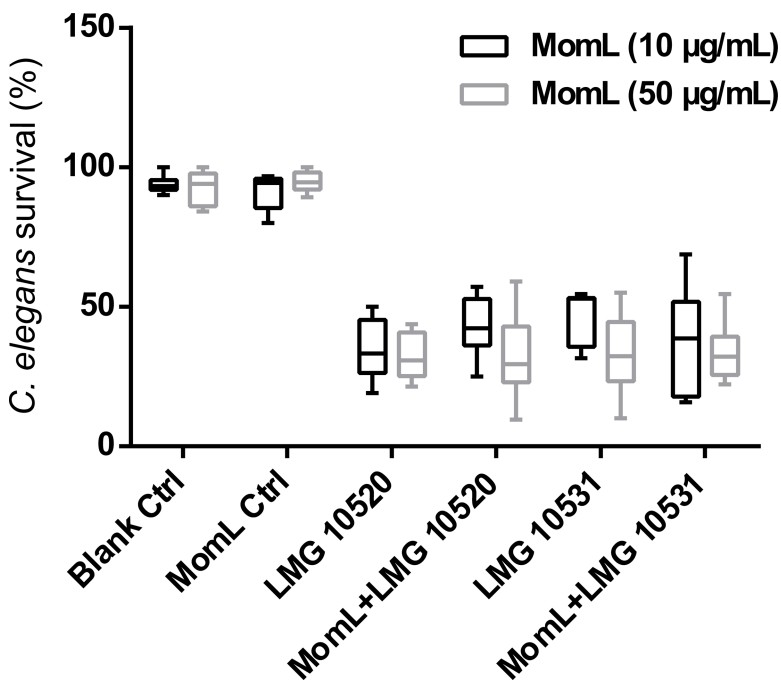

**Figure 10** **Effect of MomL on the virulence of *A. baumanii* strains in *C. elegans*.** Percent survival of *C. elegans* infected by *A. baumannii* LMG 10520 and LMG 10531. Data shown in box-whisker plots are from three biological replicates with three technical replicates each ($n = 9$). Boxes span the interquartile range; the line within each box denotes the median, and whiskers indicate the minimum and maximum values. One-way ANOVA was performed, and no significant differences were found between control and MomL treatment in both uninfected *C. elegans* and those infected by *A. baumannii* strains.

formed in 96-well microtiter plates, MomL did not potentiate the activity of TOB in this model system. Possible explanations for this are that component(s) present in this wound biofilm model protect AHL from degradation and/or interfere with the activity of MomL (potentially through interactions with proteins in the plasma), or that QS is not essential for biofilm formation and/or resistance in these conditions. Further experiments will be required to clarify this. Although MomL showed strong activity against 3-OH-C12-HSL in the medium used in the *C. elegans* model (Fig. S2), no effect of MomL on the virulence of *A. baumanii* was observed. As previously reported, an *A. baumannii* QS mutant did not differ from the wild type with regards to killing in a *Galleria mellonella* infection model (*Peleg et al., 2009*). These results indicated that although QS is known to play an important role in *A. baumanii* biofilm formation, it might only have limited role in the virulence in the *C. elegans* and *G. mellonella* models.

Thus far, a series of promising results about *in vivo* application of QQ enzymes have been reported. Phosphotriesterase-like lactonase *Sso*Pox-I has been reported to reduce biofilm formation of *P. aeruginosa* at a concentration higher than 170 µg/mL, and the early use of *Sso*Pox-I reduced the mortality of rats with acute pneumonia from 75% to 20% (*Hraiech et al., 2014*). In another study, acylase-containing coatings on silicone urinary catheters reduced formation of *P. aeruginosa* biofilms and mixed-species *P. aeruginosa-E. coli* biofilms (*Ivanova et al., 2015*). Our data obtained in a dual-species biofilm formed by *P. aeruginosa*

and *A. baumannii* as well as in a wound model strongly suggest that the effect of MomL (and potentially also other QQ enzymes) on *in vivo* grown bacterial biofilms may be much less pronounced than the effect observed with biofilms formed under simple *in vitro* conditions. Factors affecting the anti-biofilm activity in more complex systems could include stability of the enzyme, penetration of the enzyme through the biofilm matrix, and the composition of the environment.

Different outcomes were also observed when we evaluated the effect of MomL on different *Acinetobacter* strains, and no effects of MomL on biofilm formation was detected for three out of five *Acinetobacter* strains tested. In addition, for *A. baumanii* LMG 10520, a considerably higher concentration of MomL was required to obtain a pronounced inhibitory effect than for *A. baumannii* LMG 10531. These results confirm that the anti-biofilm activity of QQ enzymes is strain-dependent, which is likely to reduce their clinical efficacy.

## CONCLUSION

The results of the present study highlight that there are considerable hurdles to be cleared before QQ enzymes could potentially be used to combat infections. Our data indicate that demonstrating AHL degrading activity *in vitro* and/or anti-biofilm activity in simple *in vitro* biofilm model systems is not sufficient to predict an anti-biofilm effect in more complex systems.

## ACKNOWLEDGEMENTS

We thank prof. Xiao-Hua Zhang for providing *Escherichia coli* BL21(DE3) harboring the MomL expression plasmid pET24a(+)-momL-(−SP), prof. Wim Quax for providing *A. nosocomialis* M2 and prof. Colin Manoil for providing *A. baumannii* AB5075.

### Funding

This work was supported by the China Scholarschip Counil (CSC) and the Special Research Fund of Ghent University. The funders had no role in study design, data collection and analysis, decision to publish, or preparation of the manuscript.

### Grant Disclosures

The following grant information was disclosed by the authors:
China Scholarschip Counil (CSC).
Special Research Fund of Ghent University.

### Competing Interests

Tom Coenye is an Academic Editor for PeerJ.

### Author Contributions

- Yunhui Zhang conceived and designed the experiments, performed the experiments, analyzed the data, wrote the paper, prepared figures and/or tables.

- Gilles Brackman conceived and designed the experiments, analyzed the data, wrote the paper, prepared figures and/or tables, reviewed drafts of the paper.
- Tom Coenye conceived and designed the experiments, wrote the paper, prepared figures and/or tables, reviewed drafts of the paper.

### Data Availability

The raw data has been supplied as Supplementary File.

### Supplemental Information

Supplemental information for this article can be found online at http://dx.doi.org/10.7717/peerj.3251#supplemental-information.

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
