# Peer review of "Pitfalls associated with evaluating enzymatic quorum quenching activity: the case of MomL and its effect on Pseudomonas aeruginosa and Acinetobacter baumannii biofilms"

_PeerJ, doi:10.7717/peerj.3251_

## Round 0.1 · original submission · Major Revisions

In addition to the comments of the reviewers, I would like you to consider presenting your data in a more appropriate way. It is very common to see quantitative data like yours presented as bar charts and analysed using a Student’s t-test. But neither of these are appropriate.

T tests are for comparing two groups only. If you select p=0.05 then you have a 5% chance of a false positive (known as a type 1 error [http://bit.ly/2koxRBV]). The more t tests you do, the more chance of a false positive. To minimise this, there are other tests that are done when there is more than one group to compare, called analyses of variance tests or ANOVAs. These tests correct for the number of tests performed to minimise the chance of you making a type 1 error.

There are also different tests depending on whether your data is normally distributed or not. You need to check this first by running a normality test. I use the GraphPad Prism graphing and stats package as this guides you through the appropriate analyses. If you don’t have access to this package or something similar, you need to see someone who can advise you on statistical analysis.

Figure 1: You say that this data represents the average of n=3. Is this 3 biological or technical repeats? For that small a number of samples, please present as dot plots of the individual data points, with a line to indicate the median. As you have more than one group to compare, and the data is unlikely to be normally distributed, data should be analysed using a non-parametric version of an ANOVA with posthoc tests to compare the columns you are interested in comparing.

For the rest of your figures: Are the averages biological or technical repeats? This information needs to be in the legend. If they are biological repeats, and n=>8, best to present as box-whisker plots (http://bit.ly/2kQ4hsi). Otherwise present as dot plots. Test for normality and then run the appropriate ANOVA with posthoc tests to compare the columns you want to compare.

Reviewer 1 ·

Basic reporting

Overall, the manuscript is well-written and clear.

I know it isn't a criteria for PeerJ, but I'd like to add that this research addresses both an interesting and important problem (at least in my opinion). I enjoyed reading this manuscript.

However, I do have a few concerns and points that need to be addressed before it is suitable for publication.

-For the crystal violet assays, it would be better to show the absolute/raw data, rather than normalised (relative to the control) data. As presented, there is no way for the reader to assess whether or not a significant amount of biofilm was actually formed (e.g. I would place more significance on the data if the absorbance of the positive/untreated control was 0.4, rather than 0.04, for example).

-It should be clarified in the introduction what the the activity of MomL is towards the two AHL signalling molecules used by Pseudomonas aeruginosa (N-(3-oxododecanoyl)-L-homoserine lactone (3-oxo-C12-HSL) and N-butyryl-L-homoserine lactone (C4-HSL)). Is it active towards both? If so, are the activity levels similar towards both?

If it is only active against C4-AHSL, this may also tie into the discussion (i.e. the relatively poor effect on biofilm formation could be due to low activity towards 3-oxo-C12..?). This needs to be clarified and/or discussed.

-It is mentioned in the intro that 'other AHLs with varied chain lengths..are also found in Acinetobacter strains" - is it known what AHLs are used by the specific strains tested as part of this work?

Experimental design

- Given that MomL has ~50% activity at pH 6 (as compared to pH 7) why is it stored in a pH 6.5 buffer? And with no salt? Is there any data available that shows the enzyme is 100% active in this buffer (Especially since this is a different buffer than used in the previous work)? Or might the poor results in the biofilm assays be because the enzyme is partially inactivated under these buffer/storage conditions?

- When using the microtiter plate assay, it is important to have a negative control (e.g., uninoculated medium) on each test plate. Was this control performed?

Validity of the findings

-For all data presented, it needs to be clarified whether the replicates are technical replicates, or biological replicates/independent experiments.

- Why does the addition of protein alone reduce the number of cells by ~50%? Is this expected? Is this specific to MomL? Or would any protein have this effect? Perhaps more importantly, could this growth defect be being mistaken for a lack of attachment capability in the crystal violet assays?!

-Why do some experiments have 6 replicates, some 9 replicates, and some >27? What this part of the original experimental plan? I'm guessing it is something to do with plate layout/space available on the 96-well plates, but it seems odd (and it is best practice to decide in advance how how many replicates will be performed, in order to avoid inadvertent p-hacking).

Additional comments

- What the controls are in the various figures should be made more explicit. In particular, in Figure 3, I assume the control is 'no enzyme' (the other figures at least mention 'untreated control').

- it is mentioned that abaI mutants have ~40% reduced biofilm formation. This is a very useful benchmark! (i.e. this is the maximum effect one would expect from a QQ enzyme on Acinetobacter biofilms). Are similar numbers available for lasR/Rhl deletions in Pseudomonas? If so, it would be good to include them. If deletion of these enzymes (or other QQ strategies) show a greater effect than is observed with MomL - it changes the interpretation of the data quite significantly. It may not be that ''less advanced" biofilm assays are misleading, but rather that the MomL enzyme isn't very good at disrupting P. aeruginosa biofilms. The effect of MomL on P. aeruginosa biofilms in the crystal violet assays seems relatively weak (only a ~35% reduction, with very high enzyme concentrations required - and this may be largely due to it killing the cells!). Therefore, I'm not yet convinced that "all these in vitro results seem promising", and it doesn't seem that surprising that the results in subsequent biofilm assays are also underwhelming. Perhaps something to expand upon/clarify...

- Very minor points: Figure 1 - How was the gel stained? Regular Coomassie? It looks a bit odd/faint/pixelated. Also, is the molecular mass of 'nearly 31kD', the expected molecular mass? (And shouldn't it be kDa, not kD?).

Reviewer 2 ·

Basic reporting

The authors utilize clear, well written English throughout the manuscript to tell a robust and well defined scientific story on quorum quenching and biofilms. What seems to be lacking is a careful combing of the literature to compare what the authors found and has been published already and highlighting this in the literature. For instance claiming that "To our knowledge, this is the first study to evaluate the anti-biofilm activity of a QQ enzyme in
more advanced biofilm models (including multispecies biofilms and a wound biofilm model)." is very bold when there are at least of handful of studies from a quick search on pubmed that come up which have used QQ enzymes against multispecies and in vivo biofilms.

e.g. Quorum-Quenching and Matrix-Degrading Enzymes in Multilayer Coatings Synergistically Prevent Bacterial Biofilm Formation on Urinary Catheters.
https://www.ncbi.nlm.nih.gov/pubmed/26593217

Also need to do a better job relating the historical studies of MomL in the discussion to help the reader fully understand the story. Also what about the use of host components with other enzymes and how they can impede their use? Overall the authors need to include more comparative literature references.

Experimental design

Experimental design seems thoughtful and well planned. Only idea is to look at a mechanism of why it's not working in the presence of in vivo host components? Maybe test certain host components of the wound biofilm to see which one is inhibiting the QQ enzyme's activity; for sure plasma with all the binding proteins? Not needed to publish but should discuss this in the discussion setting. Also where are the authors going from here? Trying to find a mechanism of action for why its not working, going to test other enzymes, etc?

Validity of the findings

See above sections but the authors prepared an informative publication which will be very helpful to others in the literature. Not ground breaking but essential and important work.

Additional comments

Nice work, I enjoyed reading your paper and the diverse ways you came to attack the problem and still maintaining host relevance. Kudos to you and your lab!!

·

Basic reporting

The article is professionally written. I am satisfied with the referencing and background provided to the work. All article sections are presented. In my opinion, the Discussion section should be made more comprehensive by expanding on the interpretation and caveats of the observed results. I have outlined my questions and comments in the 'General comments for the author' section below. Addressing these points in the discussion will help provide a more complete picture of this work.

Raw data is appropriately labeled for the convenience of the readers. However, I did notice some typos/ambiguities in the text:

Line 19: evaluate should be evaluated
Lines 20 and 21: baumanii should be baumannii
Line 22: MomL ‘increased’ biofilm susceptibility
Line 46: What are the molecular targets of furanones?
Line 50: Do you mean the peptidoglycan biosynthesis machinery (ref: your Sci Rep 2016 paper)?
Line 255: sentence formation (which indicates the presence of other mechanisms regulating biofilm formation)
Line 263: In 'the' wound model... ('this' should be replaced by 'the')

I request the authors to address these in the resubmission.

Experimental design

This work will inform the research community, especially research groups working on biofilm models of antibiotic efficacy and resistance, on the caveats to be considered while designing experiments and interpreting results. Therefore, the research aligns well with the Aims and Scope of PeerJ.

The assays employed in this work have been used previously by the authors in research that has been published in reputable, peer-reviewed journals. Appropriate controls, biological replicates and significance testing has been conducted to validate the results.

Validity of the findings

No comments. See general comments for the author below.

Additional comments

Antibiotic resistance is an important health concern and poses serious challenges to the research community. Conventional antibiotics are no longer effective, therefore, novel strategies to counter the spread of multi-drug resistant bacteria are needed. Quorum sensing inhibitors (QSIs) and Quorum quenching (QQ) enzymes are promising candidates in the fight against drug resistant infections. While bacteria can exist in planktonic state as well, they are more resistant to drugs in their relatively inert forms such as spores and biofilms, which are also more commonly encountered in real life settings. This manuscript examines the biofilm inhibitory activity of a QQ enzyme (MomL) in various environments. Enzyme activity was assayed via the liquid X-gal assay in a broth based bacterial growth medium. This behooves the question of enzyme activity in the advanced model environments studied in this work. Biofilm formation is assayed via crystal violet staining, plating and CFU counts, as well as microscopy. Assay conditions include bacterial growth media (MH broth), wound model and a C elegans culture. MomL-mediated biofilm inhibition and susceptibility to conventional antibiotics were diminished in the wound model and C elegans, as compared to the standard, in vitro conditions. These findings could guide researchers working in this area to improve experimental design and assay conditions.

I request the authors to consider the following questions and comments. Many of these points can be addressed by supplementing the discussion section appropriately.

1) In all the biofilm formation work described in this manuscript, an overnight bacterial culture is diluted and added to microtiter wells. At this time, are the cells in stationary phase or experiencing vegetative growth? Next, MomL is added to this cell suspension, followed by incubation. Since MomL is added to a cell suspension, I am curious if the effects of MomL on planktonic bacteria are known or have been studied by the authors? If yes, please reference those works, and include a brief discussion.

2) How do the data in Figures 2 and 3 compare? In figure 2, 1 microgram/mL MomL results in near-complete degradation of the AHL. In biofilms, this concentration has negligible effects in P aeruginosa and only ~40% effect in A baumannii biofilms. Why is MomL more potent in inhibiting A baumannii biofilms (0.1 microgram/mL) than P aeruginosa biofilms (50 microgram/mL)?

3) The authors have quantified biofilm formation using two methods - crystal violet staining (Figure 3) and determination of CFUs/biofilm (Figure 4). Do these two assays report similarly on biofilm abundance? It would be helpful to the readers if the authors provide a comparison of data from both assays in a single experiment, or refer to previously published work describing the same. Also, a discussion on the strengths and weaknesses of the two methods to quantify biofilms will guide readers interested in adopting these biofilm assays in their own research.

4) The biofilm abundance in the control (no MomL) for P aeruginosa (2.366) is more than two-fold higher than that for A baumannii (1.097) (supporting info Sheet 2). Is this a legitimate comparison, i.e. were the assays run on the same plate/instrument/day? According to the authors, what are the factors responsible for this difference - biological differences in biofilm size/proliferation rates of the two species? Or differential staining of the cells by crystal violet?

5) Looking at the control data from the susceptibility assay (raw data, Sheet 3), the CFU/biofilm values for both species are comparable. Could the authors explain this, especially in the context of question 4 above?

6) Which steps of biofilm formation does MomL inhibit? Nucleation of the biofilm? Biofilm proliferation?

7) Fluorescence microscopy:

a) I urge the authors to describe their fluorescence images in greater detail.
b) Please include an image of a blank well, as control.
c) Brightness and contrast should be scaled equally for all images within a panel.
d) In Methods, please provide details of the image acquisition settings, such as the objective numerical aperture and magnification, exposure times, excitation and emission wavelengths/bandwidths, details of monochromator or emission filter-dichroic systems used.
e) How were the images analyzed – which software was used, was there any normalization or background subtraction performed on the raw data?

8) Dual species biofilm formation:

a) Please explain your choice of starting cell densities for P. aeruginosa (5x10^5 CFU/ml) and A. baumannii (5x10^7 CFU/ml).
b) I am curious about the composition of the biofilm formed from a mixture of P. aeruginosa and A. baumannii suspensions. Has the ‘dual composition’ of the resulting biofilm been confirmed? Can we rule out the possibility that two distinct biofilms exist, one per species? Or only one ‘single-species biofilm’ due to inhibition of the other species’ biofilm formation, as observed in this work? Are there any estimates of rate of biofilm growth for the two species in separate biofilms?
c) The data shows that P aeruginosa inhibited growth A baumanni in the ‘dual species’ biofilm assay in the control sample (zero MomL). This particular combination of bacteria may be unsuitable for studying inhibitory effects of MomL on mixed species biofilms. I am not convinced that this figure adds substantially to the main text. I suggest the authors to consider providing this figure and related text in Supporting Information.

9) Biofilm formation in wound model: Have the authors previously confirmed biofilm formation in the media used in this assay?

10) C elegans assay:

a) Is killing of worms carried out by biofilms or by vegetatively growing bacteria or both? Which of these are active in this work?
b) Similar to question 9, has biofilm formation and MomL activity on biofilms been confirmed in the medium used in this assay (in the absence of the worms)?
c) Have the authors considered sampling a range of variables such as MomL concentrations, incubation times and A. baumannii starting concentrations? In my opinion, this would be a useful approach to conclusively rule out MomL mediated biofilm inhibition in a real-life environment.
d) The MomL conc used in this assay was 10 microgram/mL, for both A baumannii strains. However, data for LMG10520 in figure 7 and the text (line 222) suggest that at least 50 microgram/mL of MomL should be used to significantly inhibit biofim formation in this species. Even at 10 microgram/mL, there is a small increase in worm survival, although the error bars are wide. It might be worthwhile to repeat this assay for LMG10520 with 50 microgram/mL or higher concentrations of MomL.
e) There is no discussion of the results from this assay in the Discussion section.

11) To test whether loss of enzyme stability or activity contributes to the lack of biofilm inhibition in advanced models (wound model and C elegans), would it be worthwhile to assay for MomL activity (liquid X-gal assay) under the same media conditions?

---

## Round 0.2 · Minor Revisions

As I indicated in my previous review, you should present the data that you have left as bar charts as box-whisker plots. These give a much better representation of the variability of your data.

---

## Round 0.3 · accepted · Accept

I'm glad I've managed to convert you to using box-whiskers! They are so much more informative than bar charts.